# Generalization of ALMM Based Learning Method for Planning and Scheduling

Zbigniew Gomolka [1],* , Ewa Dudek-Dyduch [2] and Ewa Zeslawska [1]

1    College of Natural Sciences, University of Rzeszow, Pigonia St. 1, 35-959 Rzeszow, Poland
2    Faculty of Electrical Engineering, Automatics, Computer Science and Biomedical Engineering,
     AGH University of Science and Technology, 30-010 Cracow, Poland
*    Correspondence: zgomolka@ur.edu.pl

**Abstract:** This paper refers to a machine learning method for solving NP-hard discrete optimization problems, especially planning and scheduling. The method utilizes a special multistage decision process modeling paradigm referred to as the Algebraic Logical Metamodel based learning methods of Multistage Decision Processes (ALMM). Hence, the name of the presented method is the ALMM Based Learning method. This learning method utilizes a specifically built local multicriterion optimization problem that is solved by means of scalarization. This paper describes both the development of such local optimization problems and the concept of the learning process with the fractional derivative mechanism itself. It includes proofs of theorems showing that the ALMM Based Learning method can be defined for a much broader problem class than initially assumed. This significantly extends the range of the prime learning method applications. New generalizations for the prime ALMM Based Learning method, as well as some essential comments on a comparison of Reinforcement Learning with the ALMM Based Learning, are also presented.

**Keywords:** machine learning; reinforcement learning; metaheuristic; optimization method; discrete optimization; scheduling; multistage decision process; heuristic methods; ALMM

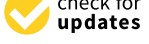



## 1. Introduction

Curiosity is the main motivation for any human, a researcher in particular. Whenever a new method or algorithm is developed, multiple questions emerge. Could the method be applied under weaker assumptions, thus enhancing the area for its use? Could it be improved? In which ways? How does the method relate to the ones that came earlier? This paper refers to a learning method (a metaheuristic) called the ALMM Based Learning method. The objective of the method is to solve discrete (combinatorial) optimization problems, especially NP-hard ones. This paper presents further results of research involving the method and thus provides answers to the questions above. The method utilizes a special multistage decision process modeling paradigm referred to as the Algebraic Logical Metamodel of Multistage Decision Processes (abbreviated as ALMM of MDP and finally ALMM), which is presented in Section II. Based on the ALMM paradigm, one can develop mathematical models for discrete optimization problems—the so-called AL models. Even though a problem model may be known, its analytic solution is not available. Moreover, as the model has a recursive character, it is difficult to infer decision consequences by more than a single step. Initial work related to attempts to use algebraic logic models to optimize discrete process control have been proposed in [1,2]. They used the concept of a simplified algebraic-logical description of the properties of the process that could be used to control its course. This approach had a strong limitation because it did not define a mathematical model of the process, but only relationships of its properties that could be used to make decisions controlling it [3–5]. In this paper, the authors define a generalized mathematical notation of the process model described in the ALMM technology, which takes into account the new mechanism of the learning strategy.

The objective of this paper is twofold:

- To present new generalizations of ALMM Based Learning methods involving a fractional derivative mechanism;
- To present essential remarks related to a comparison of Reinforcement Learning vs the ALMM Based Learning method.

Many types of learning have been explored for some discrete optimization problems [6–14], especially scheduling: rote learning, inductive learning, neural network learning, case-based learning, classifier systems, and others. Particular methodologies offer positive and negative features. However, none of the mentioned learning concepts use a mathematical model of a problem to be solved. The novelty of the machine learning presented in this paper is the fact that this method is based on a special mathematical model. The paper is constructed as follows. The Algebraic Logical Metamodel of Multistage Decision Process is presented in Section 2. Section 3 presents definitions of some criterion properties that are used in further discussion. An example that is an extension of the proposed learning method strategy is recalled in Section 4. Generalizations for the prime ALMM Based Learning method, as well as essential comments on a comparison of Reinforcement Learning with ALMM Based Learning, are given, respectively, in Sections 5 and 7. In Section 6, an example is presented.

## 2. Algebraic-Logical Meta-Model of Multistage Decision Processes

The discrete process, which is implemented in the form of a sequence of control decisions, is an effective and practical approach that allows for monitoring various production processes. The key problem in constructing models of such process control is the need to have expert knowledge regarding the nature of such a process—whether it is deterministic or stochastic—but also the degree of their adjustment to real systems and the related confidence level of the conducted steering of this system. In this paper, the authors propose a mechanism of abstract mathematical modeling to describe a deterministic process using the approach paradigm named ALMM of MDP (abbreviated as ALMM). The idea of the ALMM paradigm was proposed and developed by Dudek-Dyduch E. in [1] and recalled in many other papers [5,15,16]. It has also been put to use in multiple cases [1,17–22]. Based on ALMM, the formal models, the so-called AL models, may be established for a very broad class of discrete optimization problems from a variety of application areas (especially for the modeling and control of discrete manufacturing and logistics processes), thus yielding to the meta-model designation. The description and definition of the ALMM paradigm, cited in the aforementioned articles, are cited below. "ALMM is a general model development paradigm for deterministic problems, for which solutions can be presented as a sequence (or a set) of decisions, usually complex ones (i.e., composed of some single decisions). It facilitates convenient representation of all kinds of information regarding the problem to be solved, in particular the information defining a structure of states and decisions, algorithm used to generate consecutive states and various temporal relationships and restrictions of the problem. Furthermore, ALMM enables us to define various problem properties, in particular ones for which a particular heuristic method may be applied". ALMM provides a structured way of recording knowledge of the goal and all relevant restrictions that exist within the problems modeled. Using this paradigm, the author has provided, i.e., in [3,16], the definitions of two base types of multistage decision processes: a common process (cMDP) and a dynamic process (MDDP). The definition of MDDP quoted below refers to processes wherein both the constraints and the transition function depend on time. Therefore, the concept of the so-called "generalized state" has been introduced, defined as a pair containing both the state and the time instant.

**Definition 1.** *The multistage dynamic decision process is a process that is specified by the sextuple* $MDDP = (U, S, s_0, f, S_N, S_G)$, *where* $U$ *is a set of decisions,* $S = X \times T$ *is a set of generalized states,* $X$ *is a set of proper states,* $T \subset \Re + U\{0\}$ *is a subset of non-negative real numbers representing the time instants,* $f : U \times S \rightarrow S$ *is a partial function called a transition function,*

*(it does not have to be determined for all elements of the set $U \times S$), $s_0 = (x_0, t_0)$, $S_N \subset S$ and $S_G \subset S$ are, respectively, an initial generalized state, a set of not admissible generalized states, and a set of goal generalized states, i.e., the states in which we want the process to take place at the end. Subsets $S_G$ and $S_N$ are disjoint, i.e., $S_G \cap S_N = \varnothing$.*

*The transition function is defined by means of two functions, $f = (f_x, f_t)$, where $f_x$, $f_t$ determine the next state and the next time instant, respectively. It is assumed that the difference $\Delta t = f_t(u, x, t) - t$ has a value that is both finite and positive. Because not all decisions defined formally make sense in certain situations, the transition function $f$ is defined as a partial one. As a result, all limitations concerning the decisions in a given state $s$ can be defined in a convenient way by means of so-called sets of possible decisions $U_p(s)$, and defined as: $U_p(s) = \{u \in U : (u, s) \in Domf\}$.*

*The cMDP is obtained by reducing a generalized state to a proper state with a transition function $f = f_x$. For both defined types of the multistage decision processes, in the most general case, sets $U$ and $X$ may be presented as a Cartesian product $U = U^1 \times U^2 \times ... \times U^m$, $X = X^1 \times X^2 \times ... \times X^n$, i.e., $u = (u^1, u^2, ..., u^m)$, $x = (x^1, x^2, ..., x^n)$.*

*In particular, $u^i$, $i = 1, 2, ..., m$ represent separate decisions that must or may be taken simultaneously and relate to particular objects. Values of particular coordinates of a state or a decision may be names of elements (symbols) as well as some objects (e.g., finite set, sequence etc.).*

*There are no limitations imposed on a type of elements of the sets; in particular they do not have to be numerical. Thus, values of particular co-ordinates of a state or a decision may be names of elements (symbols) as well as some objects (e.g., finite set, sequence etc.). The sets $X_N, S_N, X_G, S_G$ and $U_p$ are formally defined with the use of both algebraic and logical formulae, hence the algebraic–logic model descriptor.*

The most significant characteristic, unique for the proposed ALMM of MDP paradigm, is the fact that:

- Proper state coordinates can be higher order variables;
- Decision $u$ can take a complex form, consisting of individual decisions related to various issues/objects; these individual decisions may or have to be taken or executed at the same time;
- The transition function is defined as a partial one, which allows taking into account various different restrictions on sensible decisions in different states.

Based on the meta–model recalled herein, AL models may be created for individual problems consisting of seeking admissible or optimal solutions. In the case of an admissible solution, an AL model is equivalent to a suitable multistage decision process, hence it is denoted as process **P**. An optimization problem is then denoted as a $(\mathbf{P}, \mathbf{Q})$ pair, where **Q** is a criterion. An optimization task (instance of the problem) is denoted as a $(P, Q)$, where $P$ is an instance of the process **P** and is named an individual process. At the same time, an individual process $P$ is represented by a set of its trajectories. A finite trajectory is a sequence of consecutive states from the initial state to a final state (goal, not admissible or blind one), computed by the transition function. Though trajectories may be finite or infinite, for further consideration we assume only finite ones. For $s_0$, it is assumed that no state of a trajectory, apart from the last one, may belong to the set $S_N$. Only a trajectory that ends in the set of goal states is admissible (feasible). The decision sequence determining an admissible trajectory is an admissible decision sequence. Obviously, the set of finite trajectories corresponds to the state graph of the process $P$.

## 3. Properties of Problems

The vast majority of methods and algorithms described in the literature use (with or without overt declaration) some properties of criteria that facilitate the solving process. Some of these, defined based on the ALMM paradigm, have been presented in [3,19] and then recalled in [23], are shortly recalled below. A broad class of criteria can be defined by recurrence and computed in parallel to the calculation of trajectories, the author named the said class as separable criterions (Definition 2). It is worth remembering, though, that these

are not the only criterion classes that may be used. Let us denote $P$—a fixed multistage decision process, $S^P$—a set of all states of trajectories of the process, $d(\tilde{s})$—the number of the last state of a finite trajectory $\tilde{s}$, $\tilde{U}$—a set of all decision sequences of the process $P$, $\mathfrak{R}$—a set of real numbers.

**Definition 2.** *Criterion $Q$ is separable for the process $P$ if, for every decision sequence $\tilde{u} \in, \tilde{U}_t$ can be recursively calculated as follows:*

$$
\begin{aligned}
Q_0 &= const, \; in \; particular \; Q_0 = 0 \\
Q_{i+1} &= f_Q(Q_i, u_i, s_i) \quad for \quad i = 0, \; 1, \; \dots, \; d(\tilde{s}) - 1,
\end{aligned}
\tag{1}
$$

*where $Q_i$ for $i > 0$ denotes a partial value of criterion $Q$ calculated for the $i$-th state of the considered trajectory, defined as follows:*

$$
Q_i = Q(\tilde{u}'),
\tag{2}
$$

*where $\tilde{u}' = (u_0, u_2, \dots, u_{i-1})$ is the initial part of the sequence $\tilde{u}$, $f_Q$ is some partial function $f_Q : \mathfrak{R} \times U \times S \to \mathfrak{R}$ such that:*

$$
Dom f_Q = \left\{ (a, u, s) \in \mathfrak{R} \times U \times S : s \in S^P, u \in U_p(s), a \in \mathfrak{R} \right\}.
\tag{3}
$$

Separability is a property of an algorithm which calculates a quality criterion for a sequence of decisions $\tilde{u}$, and thus for designated trajectory $\tilde{s}$. The criterion is separable if we can calculate its value for the next state of a trajectory knowing only its value in the previous state and the decision taken at that time.

Particularly useful is the property of additive separability of criterion. Let $Q$ be a separable criterion and a function $\Delta Q$ denotes a change of criterion for two consecutive states on any trajectory, i.e., $\Delta Q_i = Q_{i+1} - Q_i$.

**Definition 3.** *Separable criterion $Q$ is additive for a process $P$, iff for each trajectory $\tilde{s}$ of the process $P$ and for each $i = 0, 1, \dots, d(\tilde{s}) - 1$ occurs:*

$$
f_Q(Q_i, u_i, s_i) = Q_i + \Delta Q(u_i, s_i),
\tag{4}
$$

*i.e., $\Delta Q$ depends on the state and decision only.*

**Definition 4.** *Separable criterion $Q$ changes multiplicatively for a process $P$, iff for each trajectory $\tilde{s}$ of the process $P$ and for each $i = 0, 1, \dots, d(\tilde{s}) - 1$ the following is true:*

$$
f_Q(Q_i, u_i, s_i) = Q_i \cdot v(u_i, s_i),
\tag{5}
$$

*where $v(u, s)$ is a certain function depending on the decision and the state only [16].*

## 4. Machine Learning Based on ALMM

In the work [3], the authors proposed a machine learning method that used the changes dynamics of the $\Delta Q$ criterion and the weighted sum of partial criteria to determine the local quality criterion $q$. The learning process was reduced to determining a set of trajectories of process states leading to $S_G$ states and sequentially searching for such control sequences $u$, taking into account the selection of the weight vector $a$. To avoid unnecessary redundancy, we recall briefly the basic form of the local criterion $q(u, s)$ which became the following form:

$$
q(u, s) = a_0 \big( \Delta Q(u, s) + \hat{Q}(u, s) \big) + a_1 \varphi_1(u, s) + \cdots + a_i \varphi_i(u, s) + \cdots + a_n \varphi_n(u, s),
\tag{6}
$$

where $\Delta Q(u, s)$ is the change of the criterion value as a result of decision $u$, undertaken in the state $s$, $\hat{Q}(u, s)$ is the estimation of the quality index value for the final trajectory section

after the decision $u$ has been realized, $\psi(u, s)$ is the components reflecting additional limitations or additional requirements in the space of states, $i = 1, 2, \ldots, n$, $a_i$ is the coefficient which defines the weight of $i$-th component $\varphi_i(u, s)$ in the criterion $q(u, s)$. The sum of weight coefficients $a_i$ for $i = 1, 2, \ldots, n$ is equal to 1.

A role of the criterion components connected with considered subsets should be strengthened for the next trajectory, i.e., the weights (priorities) of these components increase. When the generated trajectory is admissible, the role of the components responsible for the trajectory quality can be strengthened, i.e., their weights can be increased. Based on the gained information, the local optimization task is being improved during simulation experiments. This process is treated as a learning or intelligent searching algorithm. As the $q$ criterion coefficients change in consecutive iterations, the criterion assumes the following form:

$$q^{(k)}(u, s) = a_0^{(k)}\left(\Delta Q(u, s) + \widehat{Q}(u, s)\right) + a_1^{(k)}\varphi_1(u, s) + \cdots + a_i^{(k)}\varphi_i(u, s) + \cdots + a_n^{(k)}\varphi_n(u, s), \tag{7}$$

where $k$ denotes the number of iteration.

Typically, the same initial coefficient values are assumed. Changes to the $a_2, a_3, \ldots, a_n$ coefficients depend on real maximum distances of the last generated trajectory from the reachable parts of the not-admissible state set $S_N$, and the remaining special sets.

In the work [24], the fractional back propagation algorithm was proposed (FBP), which uses the fractional order derivative according to Grünwald–Letnikov fractional derivative (GL) theory. Assuming the general form definition of the integer derivative and the fractional derivative, the GL derivative can be written:

$$_{u_0, s_0}\Delta_{u,s}^{v}Q(u, s) = \lim_{h \to 0} \frac{1}{h^v} \sum_{n=0}^{[(u - u_0, s - s_0/h)]} (-1)^n \binom{v}{n} Q(u_i - u_{i-nh}, s_i - s_{i-nh}), \tag{8}$$

where $\binom{v}{n}$ denotes the Newton binomial, $v$ the order of the fractional derivative of basis function $\varphi(u, s)$, $u_0$, $s_0$ the interval range, and $h$ denotes the number of steps in the state space of the given process. Then, we can assume the backwards difference of the fractional order as $_{u_0, s_0}\Delta_{u,s}^{(v)}$, where $v \in \mathbb{R}^+$:

$$_{u_0, s_0}\Delta_{u,s}^{(v)} = \sum_{n=0}^{\lfloor (u_i - u_0, s_i - s_0)/h \rfloor} b_n^{(v)}Q(u_i - u_{i-nh}, s_i - s_{i-nh}), \tag{9}$$

where the consecutive coefficients $b_n^{(v)}$ are defined as follows:

$$b_n^{(v)} = \begin{cases} 1 & \text{for } n = 0 \\ (-1)^n \frac{v(v-1)(v-2)\ldots(v-n+1)}{n!} & \text{for } n = 1, 2, 3, \ldots, N. \end{cases} \tag{10}$$

$N$ denotes the number of discrete measurements of $Q(u, s)$. The $\Delta Q(u, s)$ in Equation (7) is therefore true for the special case when $v = 1$ and $N = 1$. Taking into account the above, it is possible to formulate a generalized form of the considered relationship, in which the change of the criterion $\Delta Q$ can be estimated by an approximation of the fractional derivative of the $v$ order in the following form:

$$q^k(u, s) = a_0^{(k)}\left(_{u_0, s_0}\Delta_{u,s}^{(v)}Q(u, s) + \hat{Q}(u, s)\right) + a_1^{(k)}\varphi_1(u, s) + \cdots + a_i^{(k)}\varphi_i(u, s) + \cdots + a_n^{(k)}\varphi_n(u, s). \tag{11}$$

Thus, in the formula adopted above, it becomes possible to search for a set of trajectories taking into account the variable length of the state vector history. In the adopted formula for determining successive coefficients implementing the approximation of the fractional-order derivative, it should be remembered that the accuracy of the approximation is determined by the length of the vector $b_n^{(v)}$. Moreover, the boundary value of $N$ for

successive sequences of process states cannot exceed the length of the generated trajectory. Using the above interpretation and the results of the research carried out for standard feed forward neural network structures using the backpropagation mechanism of the fractional-order derivative, it is possible to indicate the following features and properties of the proposed machine learning method:

- It is possible to smoothly control the fractional-order derivative approximation algorithm, using as parameters the order of the derivative $v$ and the number of vector elements $\overrightarrow{b^{(v)}}$;
- For the value of $v = 1$, the algorithm works as for the method described in [24], calculating the difference $\Delta Q$, using the property (10), such that for $N = 1$, $b_n^{(1)} = \{1, -1\}$, where $n = \langle 0, 1 \rangle$;
- For a value of v in the range of $0 < v \leqslant 2 \wedge v \neq 1$, a smooth selection of the derivative approximation is possible, allowing for the search for the trajectories of optimal solutions in the full spectrum of the space of permissible states.

The proposed algorithm is a generalization of the machine learning method proposed in [3]. The numerical stability of determining the approximation of the derivative of the $\Delta Q$ criterion using the vector $\overrightarrow{b^{(v)}}$ will depend on the number of its elements. For sufficiently large values of $N$, it is possible to determine their values in advance, also with the use of the Gamma function and further implementation in the form of a static structure, reducing the computational complexity of the algorithm at the stage of sequential search for the solution trajectory of a given optimization problem. The obtained solutions, in the form of optimal discrete process trajectories, can be used as suboptimal controls for other more complex optimization problems. At the current stage of the research, the authors will use the above machine learning model with a fractional order derivative to control the processes described in the ALMM technology to control a fleet of unmanned aero vehicles (UAV) in the conditions of dynamically changing system resources; in particular, in the case of a change in the number of UAVs, airspace availability and the intensity of the jobs stream of logistic tasks carried out by this fleet.

## 5. Generalized ALMM Based Learning Method

Generalizations of the method may take two basic directions:

- Modification of the local criterion and the learning process;
- Weakening assumptions under which the method can be applied.

### 5.1. Criterion Modifications

First of all, let us notice that for certain problems a sufficiently accurate estimation of the criterion for the remaining part of the criterion $\hat{Q}(u, s)$ may be difficult or not even possible. The learning method can be applied in such cases for the $q$ criterion, omitting the $\hat{Q}(u, s)$ addent, i.e., the formula (7) becomes:

$$q^{(k)}(u,s) = a_0^{(k)} \Delta^v Q(u,s) + a_1^{(k)} \varphi_1(u,s) + \cdots + a_i^{(k)} \varphi_i(u,s) + \cdots + a_n^{(k)} \varphi_n(u,s). \tag{12}$$

Secondly, let us notice that the reliability of $q$ criterion addends is not uniform. The $\Delta Q_i^v$ value is certain, but the other parts are merely estimates or predictions of unknown values, meaning they are not as reliable, with specifics depending on the problem instance under analysis. What is more, the reliability may even vary for individual parts of a local criterion. This suggests a deviation from the deterministic choice of the best decision for a given state, to be replaced with stochastic choice. The probability of selecting decision $u$ in state $s$ is proposed to be proportional to the $q(u, s)$ value. The proposed approach, using the probability choice, is also beneficial as the exploration of the solution space is necessary too.

### 5.2. Weakening the Assumptions

Although the learning concepts using ALMM technology, in combination with a separable additive steering quality criterion, have been discussed by the authors in previous works, we will show below that the method can also be defined for a weaker assumption, namely for a merely separable criterion.

**Theorem 1.** *Assumption: let the ALMM Based Learning method, defined for a problem* $(\mathbf{P}, \mathbf{Q})$, *be additively separable.*
*Thesis: The ALMM Based Learning method can be defined for the problem* $(\mathbf{P}, \mathbf{Q}')$, *where* $\mathbf{Q}'$ *is any separable criterion.*

**Proof.** Let us analyze the local criterion $q'(u, s)$ for a problem $(\mathbf{P}, \mathbf{Q}')$.

As restrictions represented by the process **P** remain unchanged, (and , as a consequence, the restrictions of all the instances $(P, Q')$ of the problem $(\mathbf{P}, \mathbf{Q}')$ remain unchanged), one has to consider the first two addends in formula (7) or the first addend in formula (12) only. It results from Definition 2 that:

$$\Delta^v Q = Q_{i+1} - Q_i = f_Q(Q_i, u_i, s_i) - Q_i \quad for \quad i = 0, 1, \ldots, d(\tilde{s}) - 1. \tag{13}$$

As the value of $Q_i$ is known at the state $s_i$, one can compute $f_Q(Q_i, u_i, s_i)$ and the component $\Delta Q_i$ can be used in formula (7) or formula (12). If sufficiently accurate, estimation criterion $\hat{Q}(u, s)$ may be difficult or not even possible and formulae (12) instead of (7) should be applied for the criterion $q'(u, s)$. Q.E.D.　□

**Theorem 2.** *Assumption: let the ALMM Based Learning method be defined for a problem* $(\mathbf{P}, \mathbf{Q})$.
*Thesis: The ALMM Based Learning method can be defined for the problem* $(\mathbf{P}, \mathbf{Q}')$, *where* $\mathbf{Q}'$ *is any multiplicative criterion.*

**Proof.** It results from Definition 3 that the multiplicative criterion is a separable one. Thus, the thesis resulting from Theorem 1 is true, so:

$$\Delta Q_i^v = Q_{i+1} - Q_i = Q_i \cdot v(u_i, s_i) - Q_i = Q_i \cdot (v(u_i, s_i) - 1) \quad for \quad i = 0, 1, \ldots, d(\tilde{s}) - 1 \quad Q.E.D. \tag{14}$$

These types of criteria are common in term securities investment problems that may not be redeemed early.　□

## 6. Exemplary Problem

An example of a problem that can be used as a difficult-to-control process due to the exponential complexity of the process of selecting control decisions may be a fleet of drones performing transport tasks to a network of airport destinations. Assume that the graph of a destination is a loaded undirected graph $G = (I, J, P)$ in which the set of edges denotes the air routes $J$ and the set of landing sites $I$ represent the set of nodes partially connecting the air corridors described by the relation of interconnections $P \subset (I \times J)$. This relation includes only those connections that are defined in the considered airspace structure of routes. In the considered example, it was assumed that the drones fleet used is the counterpart of a set of parallel machines, which may differ in their operating parameters. In particular, it is their cruising speed, transported weight and range. Airport tasks are characterized by due times and their deadlines. It is also assumed in the considered example that the operational time of the fleet is not subject to restrictions related to the access of airspace and the routes of individual ships can be determined independently while maintaining adequately safe air separation. The execution of the flight along a given route during the execution of the task is not subject to stoppages and individual tasks may be assigned weights, taking into account the priority of their implementation. The main task of the system is to generate the trajectory of decisions that control the process, taking into account the costs of the work of individual machines, the waiting times for assigning tasks and the total time of the execution of all tasks.

The preliminary idea of the learning algorithm for this problem was given in [2]. Other examples that refer to simplified models of discrete processes can be indicated in other studies [23,25]. The specification of individual elements of the AL model is very extensive. Below we will provide only those elements of the model that are necessary to explain the structure of the local criterion and methods of learning. The process state at any instant $t$ is defined as a vector $x = \left( x^0, x^1, x^2, \ldots, x^{jM_j} \right)$, where $M = M1 \cup M2$. A coordinate $x^0$ represents a set of corridors, namely routes to be traveled by drones to the moment $t$. The other coordinates $x^m$ describe the state of the $m$-th drone, where $m = 1, 2, \ldots, |M|$. A state $s(x, t)$ belongs to the set of not-admissible states if there is a route corridor whose job flight is not completed yet and whose due date is earlier than $t$. The definition of $S_N$ is as follows:

$$S_N = \left\{ s = (x, t) : \left( \exists j \in J, j \notin x^0 \right) \wedge d(j) < t \right\}. \tag{15}$$

A state $s(x, t)$ is a goal if all the jobs have been completed, i.e.:

$$S_G = \left\{ s = (x, t) : s \notin S_N \wedge \left( \forall j \in J, j \in x^0 \right) \right\}. \tag{16}$$

A decision determines the flights that should be started at the moment $t$, machines which are in the air, machines that should be serviced or prepared to be operational or the machines that should be waiting for weather reasons. Thus, the complex decision $u = \left( u^1, u^2, \cdots u^{|M|} \right)$, where the co-ordinate $u^m$ refers to the $m$-th drone and included all pieces of needed information. Based on the current state $s(x, t)$ and the decision $u$ taken in this state, the subsequent state $(x', t') = f(u, x, t)$ is generated by means of the transition function $f$.

In the course of trajectory generation in each state of the process, a decision is taken for which the value of the local criterion $q(u, s)$ is the lowest. The local criterion takes into account a component connected with cost of work, and a component connected with the necessity for the trajectory to omit the states of set $S_N$. The first components constitute a sum of $\Delta Q(u, x, t)$ and $\Delta \widehat{Q}(u, x, t)$, where $\Delta Q(u, x, t)$ denotes the increase of work cost as a result of realizing decision $u$ and $\Delta \widehat{Q}(u, x, t)$ is the lower estimation of the cost of finishing the set of headings matching the final section of the trajectory after the decision $u$ has been realized. As was presented in Section 4, the fractional order $\nu$ of the discrete difference $\Delta Q^\nu$ might be used to fluently scan the space of the process states.

The next component is connected with the necessity for the trajectory to omit the states of set $S_N$, i.e., it takes into account the consequences of the decision $u$ from the due date's limitations point of view. Let $L(x', t')$ estimate the minimal distance between the new state $(x', t')$ and the subset of not-admissible set $S_N$ that is reachable from the state $(x', t')$. The estimation is defined in detail in [16,26]. Thus, involving the GL fractional derivative, the local criterion is of the form:

$$q(u, x, t) = \Delta Q^\nu(u, x, t) + \widehat{Q}(u, x, t) + a \frac{1}{L(u, x', t')}, \tag{17}$$

where $a$ is a parameter that is being changed during consecutive experiments (trajectory generations). The value of criterion $q$ is computed for each $u \in U_p(x, t)$. This decision $u^*$ for which the criterion value is minimal is chosen. Then, the new state $(x, t') = f(u^*, x, t)$ is generated and the new best decision is chosen. If a newly generated trajectory is admissible and, for most of its states, the distances to the set $S_N$ are relatively big, the parameter $a$ can be decreased. In such a situation, the role of optimization compounds is enlarged. On the contrary, when the generated trajectory is not admissible, the parameter $a$ should be increased because then the greater emphasis should be put on the due date's limitations.

In a typical process of optimizing logistic tasks for unmanned aircraft, the process is affected by problems related to downtime (e.g., vessel failures or delays in flight tasks), continuous control of fleet vessels, their availability, optimal task planning, optimal use of

the available capacity, and available airspace capacity. In current air task planning systems (such as the European ATM Master Plan), this is still a manual process that is supported by additional technical means. Intelligent technology is used to control the process of executing dynamically incoming air tasks by a fleet of drones that move concurrently according to the distribution of nodal points—individual air destinations. For such a use of algebraic–logical description, the following assumptions are defined:

- The task of controlling the process of executing dynamically incoming aerial tasks by a fleet of drones that move concurrently according to the distribution of nodal points—individual aerial destinations—is defined.
- The aviation destinations for individual tasks can dynamically appear in the system, according to the properties of the process state vector.
- The drone fleet consisted of six unmanned aerial vehicles (UAV1–UAV6).
- The UAVs take off and land at a dedicated airport; the nodal points are known.
- The tasks to be executed arrive in real time and are respectively defined as $T1, T2, \ldots, Tn$, having the structure of single-stage and multi-stage target nodes, respectively.
- The execution of the assigned individual tasks for the UAV is carried out, respectively, at specific time intervals depending on the currently prevailing conditions and external interference, as well as taking into account the conditions related to possible dangerous/collision situations of the UAV.

A number of studies have been conducted for different trajectory classes, number of drones and tasks. The example presented below is for a fleet of four drones that had to perform a set of 21 random tasks for a single-stage trajectory (i.e., the number of target nodes is 1). Table 1 shows the obtained results; in the Drones column, the obtained values in the form of A × B describe the number of steps in which the tasks were completed with the initial zero state (A) and the number of drones performing the tasks (B). For the cases considered, the shortest time for the drone fleet to complete incoming tasks is 8269$[s]$, and the longest is 8507$[s]$.

**Table 1.** Results of simulations with four UAVs.

| No. | Drones | Time $[s]$ |
|---|---|---|
| 1 | 22 × 4 drone | 8269 |
| 2 | 21 × 4 drone | 8305 |
| 3 | 22 × 4 drone | 8340 |
| 4 | 22 × 4 drone | 8507 |
| 5 | 22 × 4 drone | 8321 |
| 6 | 21 × 4 drone | 8408 |
| 7 | 22 × 4 drone | 8356 |
| 8 | 22 × 4 drone | 8435 |
| 9 | 21 × 4 drone | 8279 |

Table 2 shows the trajectories in which the individual drones carried out the assigned tasks (value 0—means the completion of the task, 1–21—task numbers) for which $S_G$ was found, respectively.

**Table 2.** Assigned tasks for the UAV fleet.

| Trajectory Number | Drone 1 | Drone 2 | Drone 3 | Drone 4 |
|---|---|---|---|---|
| 1 (super-zero condition) | 0 | 0 | 0 | 0 |
| 2 | 0 | 2 | 8 | 21 |
| 3 | 0 | 11 | 16 | 21 |
| 4 | 4 | 0 | 16 | 21 |
| 5 | 4 | 1 | 0 | 21 |
| 6 | 4 | 1 | 2 | 0 |
| 7 | 0 | 1 | 2 | 18 |
| 8 | 9 | 1 | 0 | 18 |
| 9 | 9 | 0 | 3 | 18 |
| 10 | 9 | 17 | 3 | 0 |
| 11 | 9 | 17 | 0 | 12 |
| 12 | 0 | 17 | 8 | 12 |
| 13 | 20 | 0 | 8 | 12 |
| 14 | 20 | 6 | 0 | 12 |
| 15 | 20 | 6 | 15 | 0 |
| 16 | 20 | 6 | 0 | 5 |
| 17 | 0 | 6 | 14 | 5 |
| 18 | 7 | 0 | 14 | 5 |
| 19 | 7 | 13 | 14 | 0 |
| 20 | 0 | 13 | 14 | 10 |
| 21 | 0 | 0 | 0 | 10 |
| 22 | 0 | 0 | 0 | 0 |

Figure 1 shows the next steps in the tasks performed by the drone fleet.

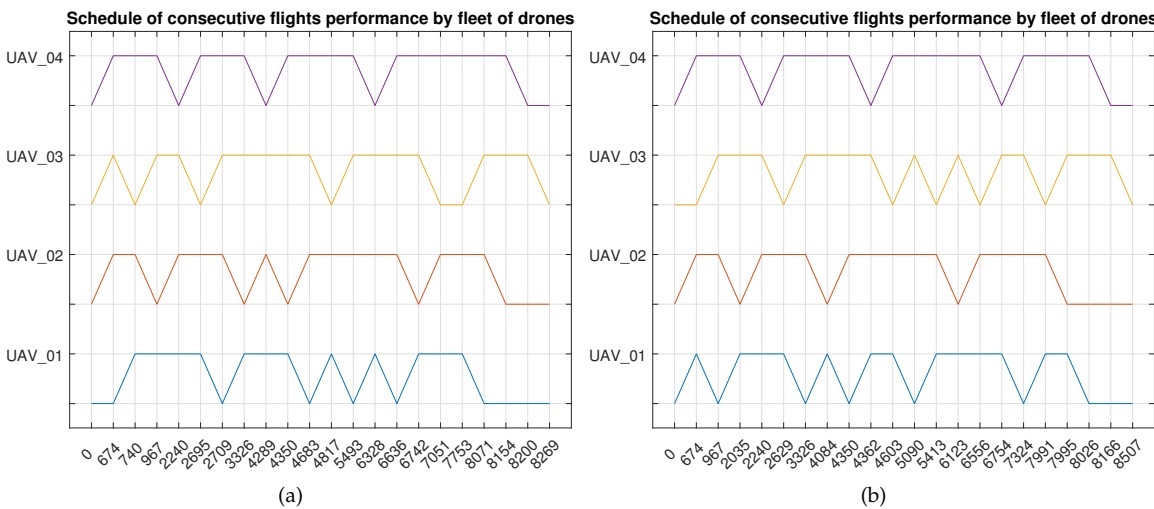

(a)  (b)

**Figure 1.** Schedule of consecutive flights' performance by fleet of four drones. (**a**) the shortest and (**b**) the longest time.

Further simulations were carried out for a randomly determined fleet of seven drones and 29 tasks, in which the solution with the shortest time of 6401 [*s*] and the longest time of 7199 [*s*], respectively, was obtained (see Table 3).

**Table 3.** Results of simulations with four UAVs.

| No. | Drones | Time [*s*] |
|---|---|---|
| 1 | 30 × 7 drone | 6911 |
| 2 | 30 × 7 drone | 6676 |
| 3 | 30 × 7 drone | 6729 |
| 4 | 30 × 7 drone | 6664 |
| 5 | 30 × 7 drone | 6708 |
| 6 | 30 × 7 drone | 6546 |
| 7 | 30 × 7 drone | 6948 |
| 8 | 30 × 7 drone | 6940 |
| 9 | 30 × 7 drone | 6471 |
| 10 | 30 × 7 drone | 6929 |
| 11 | 29 × 7 drone | 7101 |
| 12 | 30 × 7 drone | 6893 |
| 13 | 30 × 7 drone | 6895 |
| 14 | 30 × 7 drone | 6719 |
| 15 | 30 × 7 drone | 6546 |
| 16 | 30 × 7 drone | 6991 |
| 17 | 30 × 7 drone | 6918 |
| 18 | 30 × 7 drone | 6918 |
| 19 | 30 × 7 drone | 6848 |
| 20 | 30 × 7 drone | 6757 |
| 21 | 30 × 7 drone | 6487 |
| 22 | 30 × 7 drone | 6914 |
| 23 | 30 × 7 drone | 6626 |
| 24 | 30 × 7 drone | 6465 |
| 25 | 30 × 7 drone | 6879 |
| 26 | 30 × 7 drone | 6940 |
| 27 | 30 × 7 drone | 7199 |
| 28 | 30 × 7 drone | 6401 |
| 29 | 30 × 7 drone | 6982 |

Table 4 shows the assigned tasks for the drone fleet for which a target solution has been received (value 0–task completion, 1–29—task numbers).

Figure 2 shows the schedule of tasks carried out by the supervised drone fleet of seven drones.

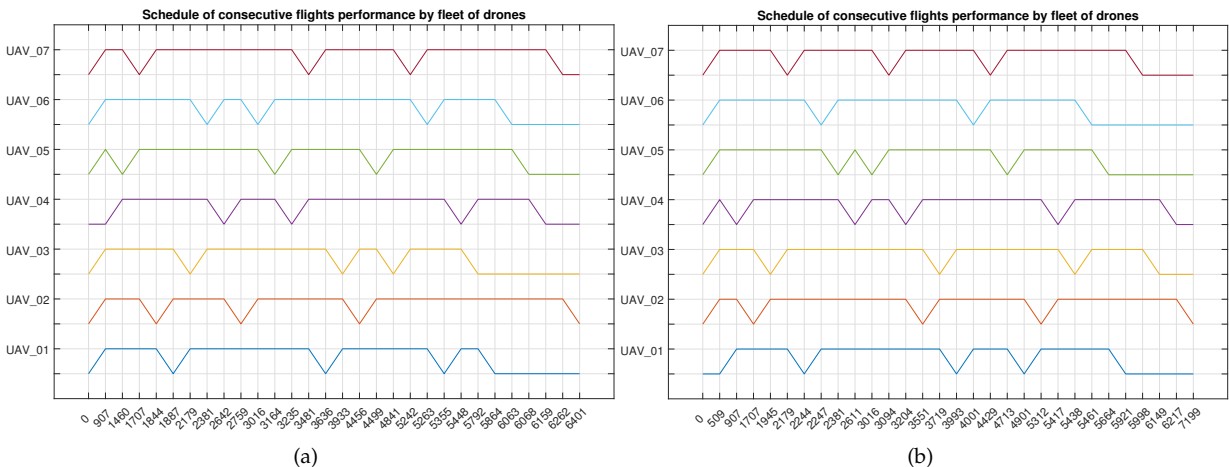

**Figure 2.** Schedule of consecutive flights performance by fleet of seven drones. (**a**) the shortest and (**b**) the longest time.

**Table 4.** Assigned tasks for the UAV fleet.

| Trajectory Number | Drone 1 | Drone 2 | Drone 3 | Drone 4 | Drone 5 | Drone 6 | Drone 7 |
|---|---|---|---|---|---|---|---|
| 1 (super-zero condition) | 0 | 0 | 0 | 0 | 0 | 0 | 0 |
| 2 | 20 | 0 | 25 | 4 | 5 | 29 | 7 |
| 3 | 20 | 0 | 25 | 4 | 5 | 29 | 7 |
| 4 | 20 | 12 | 25 | 4 | 0 | 29 | 7 |
| 5 | 20 | 12 | 25 | 4 | 10 | 0 | 7 |
| 6 | 20 | 12 | 25 | 0 | 10 | 14 | 7 |
| 7 | 0 | 12 | 25 | 26 | 10 | 14 | 7 |
| 8 | 21 | 12 | 25 | 26 | 10 | 14 | 0 |
| 9 | 21 | 12 | 0 | 26 | 10 | 14 | 8 |
| 10 | 21 | 0 | 22 | 26 | 10 | 14 | 8 |
| 11 | 21 | 15 | 22 | 0 | 10 | 14 | 8 |
| 12 | 21 | 15 | 0 | 28 | 10 | 14 | 8 |
| 13 | 21 | 0 | 18 | 28 | 10 | 14 | 8 |
| 14 | 21 | 11 | 18 | 28 | 10 | 0 | 8 |
| 15 | 0 | 11 | 18 | 28 | 10 | 13 | 8 |
| 16 | 23 | 11 | 18 | 28 | 0 | 13 | 8 |
| 17 | 23 | 11 | 18 | 28 | 2 | 13 | 0 |
| 18 | 0 | 11 | 18 | 28 | 2 | 13 | 19 |
| 19 | 3 | 0 | 18 | 28 | 2 | 13 | 19 |
| 20 | 3 | 27 | 18 | 0 | 2 | 13 | 19 |
| 21 | 3 | 27 | 0 | 24 | 2 | 13 | 19 |
| 22 | 3 | 27 | 6 | 24 | 2 | 0 | 19 |
| 23 | 3 | 27 | 6 | 24 | 2 | 17 | 0 |
| 24 | 3 | 27 | 6 | 24 | 0 | 17 | 1 |
| 25 | 3 | 0 | 6 | 24 | 9 | 17 | 1 |
| 26 | 3 | 0 | 6 | 24 | 9 | 17 | 0 |
| 27 | 0 | 0 | 6 | 24 | 9 | 17 | 0 |
| 28 | 0 | 0 | 6 | 24 | 9 | 0 | 0 |
| 29 | 0 | 0 | 6 | 0 | 9 | 0 | 0 |
| 30 | 0 | 0 | 6 | 0 | 0 | 0 | 0 |
| 31 | 0 | 0 | 0 | 0 | 0 | 0 | 0 |

The above results take into account only the obtained sets of acceptable or so-called admissible trajectories . These are only those trajectories that allow a process to reach the goal state. Unacceptable trajectories also referred to as non-admissible trajectories, which bring the process to forbbiden states, for example, as a result of over-timing the task to be executed, have been omitted. However, they can provide additional information to search for acceptable solutions to protect the process from entering a forbidden state.

## 7. Reinforcement Learning vs. ALMM Based Learning

ALMM Based Learning exhibits certain similarities to a general and much more well known method, namely Reinforcement Learning (RL) [6,8,9,11,15,27]. Let us attempt to review these similarities as well as differences. It is worth remembering, though, that these remarks do not constitute an entire comparison of the two methods. Such a comparison would not be possible anyway due to the large variety of RL method variants available. A more comprehensive comparative discussion will be the subject of a separate study. Similarities may be noticed more clearly when uniform terminology is used, which is why this paper attempts to provide equivalent terms for both methods.

### 7.1. Application of Learning

The Reinforcement method is a much more general method, which is why its applications cover multiple fields such as controlling robots, planning and scheduling problems, various games and many more. ALMM Based Learning was developed to deliver approxi-

mate solutions of NP-hard discrete optimization problems. It can be noticed, however, that the approach can be utilized for a broader spectrum of applications.

### 7.2. Environment

Both methods utilize the state space and both build the best possible trajectory (or its parts). In both cases the ultimate goal of an algorithm is to find a most profitable (overall) sequence of actions/decisions. The environment for the ALMM Based Learning method, however, is defined by the Algebraic Logical Metamodel of Multistage Decision Processes, while for the RL method it is defined as the Markov Decision Process. Thus, the ALMM Based Learning environment is deterministic, while the Reinforcement Learning one is stochastic. The next difference stems from the fact that ALMM Based Learning may utilize the so-called generalized states space $S = X \times T$, with $X$ corresponding to a space of proper states and $T$ representing the time instants space. As a consequence, ALMM Based Learning suits non-stationary environments well. All components defining the environment may be non-stationary, including the transition function $f(u,s) = f(u,x,t)$ as well as possible decision sets $U_p(s)$, (matching subsets of actions available in particular states) and constraints defining goal state set $S_G$ and not-admissible state set $S_N$.

### 7.3. Completeness of Information

ALMM Based Learning utilizes a mathematical model of a problem (or instance) to be solved. One may say it has complete information on the environment beforehand. Taking into account the recursive character of the AL models, though such information may only be utilized locally, a certain degree of analysis of the environment properties is possible; however, an estimation of the distance to the not-admissible state set and other sets is necessary. In contrast to ALMM based Learning, the RL scheme implies that there is little need for human expert knowledge about the domain of application.

### 7.4. Learning Scheme

In the RL method, the agent is supposed to choose the best action based on the current state. The selection is made based on properly built Value Functions taking various forms depending on the various RL method variants. In ALMM Based Learning, the decision (that corresponds to the RL method's 'action') is calculated based on a parametric, local criterion function $q(u,s)$. Obviously, one may express it as the Agent taking a decision based on function $q$, similar to the RL method.

### 7.5. Aim of Learning

Let us now review certain similarities to the indirect "aim" of the learning processes in both methods. First of all, both methods share a common assumption: the agent selects an action/decision that will maximize the reward in long term, not only in the immediate future. Such algorithms are known to have an infinite horizon, though in practice the heuristic philosophy is applied to a finite time range or finite trajectories (so-called episodic problems). The objective of an RL based algorithm is to find a $\prod$ policy with a maximum expected return. A policy refers to mapping that assigns some probability distribution over actions with the actions selected by the policy. The objective of an algorithm utilizing ALMM Based Learning is to find coefficients $a_2, a_3, \ldots, a_n$ for a local criterion function $q(u,s)$ in a way that would optimize the value of a global criterion $Q$. The selection of a decision in a given state is performed based on the local criterion $q$. Roughly speaking, a policy, or more precisely an action value function under policy $\prod$, corresponds to a local criterion $q$ for ALMM based Learning, while the expected reward criterion corresponds to the global criterion $Q$.

### 7.6. Learning Process

In an RL method, the learning process utilizes a "reward feedback" for the agent learning its behavior. This is known as a reinforcement signal. Exact reinforcement al-

gorithms vary depending on the RL method variant. This is the main novelty presented study, namely the learning progress performed by the ALMM Based Learning strategy. Iteration for ALMM Based Learning consists of generating a whole trajectory (admissible or not-admissible). The algorithm analyzes the whole trajectory path. Knowledge of the environment is applied to generate the most suitable local criterion. The gaining of knowledge is aggregated in a form of optimized coefficient values that determine the influence of individual criterion components. The coefficients are then improved in consecutive iterations based on the analysis of individual trajectory paths.

## 8. Conclusions

This paper presents new generalizations referring to the learning approach named the ALMM Based Learning method. The learning method is applied mainly to solve NP hard discrete optimization problems, however, the proposed approach can be used efficiently with algorithms for the broad area of applications. The paper includes proofs of theorems showing that the ALMM Based Learning method can be defined for much broader problem classes than initially assumed, including multiplicatively separable criteria. Then, it proceeds to propose and discuss essential modifications to the local criterion. The novelty of a machine learning method based on ALMM is the fact that it is uses formal algebraic–logical models of problems to be solved. Moreover, the generalization of the changing dynamics measurement of the $\Delta Q^V$ criterion was applied with the use of a fractional derivative of the GL type. This allows for fluent searching in the state space for optimal and suboptimal solutions. However, although initial knowledge is delivered it cannot be utilized easily. Thus, the additional knowledge iis acquired and gathered during successive experiments which consist of the generation of subsequent trajectories. Comparing the method with methods of machine learning [8,28–30], the presented approach has similarities to Reinforcement Learning. However, it is not its typical class. A large number of difficult problems can be efficiently solved by means of the presented method. The method is especially very useful for difficult scheduling problems with state-dependent resources. The managing of projects, especially software projects, belongs to this class. Multiple experiments were carried out, confirming the efficiency of the presented method. Simultaneously, the experiments indicated the need for further research regarding coefficient fine-tuning algorithms. More detailed classification of problems based on properties of the AL problem models is also necessary, to be followed by studies for individual classes.

**Author Contributions:** Conceptualization, Z.G., E.D.-D. and E.Z.; methodology, Z.G., E.D.-D. and E.Z.; software, Z.G., E.D.-D. and E.Z.; validation, Z.G., E.D.-D. and E.Z.; formal analysis, Z.G., E.D.-D. and E.Z.; investigation, Z.G., E.D.-D. and E.Z.; resources, Z.G., E.D.-D. and E.Z.; data curation, Z.G., E.D.-D. and E.Z.; writing–original draft preparation, Z.G., E.D.-D. and E.Z.; writing–review and editing, Z.G., E.D.-D. and E.Z.; visualization, Z.G., E.D.-D. and E.Z.; supervision, Z.G., E.D.-D. and E.Z.; project administration, Z.G., E.D.-D. and E.Z.; funding acquisition, Z.G., E.D.-D. and E.Z. All authors have read and agreed to the published version of the manuscript.

**Funding:** This research received no external funding.

**Conflicts of Interest:** The authors declare no conflict of interest.

## Abbreviations

The following abbreviations are used in this manuscript:

| | |
|---|---|
| ALMM | Algebraic Logical Metamodel of Multistage Decision Processes |
| cMDP | common multistage decision processes |
| MDDP | dynamic multistage decision process |
| $U$ | The set of decisions |
| $S$ | initial generalized state |

| | |
|---|---|
| $s_0$ | initial state |
| $f$ | transition function |
| $S_N$ | The set of not admissible generalized states |
| $S_G$ | The set of goal generalized states |
| $U_p$ | The set of possible decision |
| $X$ | The set of proper states |
| $X_N$ | The set of proper not admissible states |
| $X_G$ | The set of proper goal states |
| $\mathbf{P}$ | process |
| $(\mathbf{P}, \mathbf{Q})$ | The optimization problem |
| $\mathbf{Q}$ | critterion |
| $(P, Q)$ | The optimization task (instance of the problem) |
| $P$ | The instance of the process $\mathbf{P}$ and named an individual process |
| $S^P$ | The set of all states of trajectories of the process |
| $d(\tilde{s})$ | The number of the last state of a finite trajectory $\tilde{s}$ |
| $\tilde{U}$ | The set of all decision sequences of the process $P$, |
| $\mathfrak{R}$ | The set of real numbers. |
| $S_{A_i}$ | The state sets that are advantageous |
| $S_{AD_i}$ | The state sets that are disadvantageous |
| UAV | Unmanned Aerial Vehicle |

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
