# Peer review of "Generalization of ALMM Based Learning Method for Planning and Scheduling"

_applsci, doi:10.3390/app122412766_

Round 1

Reviewer 1 Report

The reviewed paper considers utilizing a multistage decision process modeling paradigm named ALMM. The studied topic is interesting. However, the paper only describes the theoretical parts. How about the performance of the integrated approach? If possible, please provide the computational results to support the idea of the authors. Besides, the ablation experiment should be included for evaluating the performance of the approach using different elements.

Author Response

Detailed responses for the reviewer comments are placed in the attachment.

Reviewer 2 Report

The paper presents a generalization referring to the learning approach named the ALMM-based Learning method. Then it discusses essential modifications to the local standard. However, the novelty of the machine learning method based on ALMM is deficient. Moreover, the generalization of dynamics measurement of Delta Q^v Criterion has been applied using a fractional derivative of GL type. Although the paper also points out that although initial knowledge is delivered, it cannot efficiently utilize. As for research on algorithms, I still hope that the author can provide data simulation or examples to verify the algorithm's implementation and the advantages introduced in the paper.

Author Response

(The authors gave the same response as above.)

Reviewer 3 Report

Dear authors, 

I have one major comment to make on your manuscript.

There are no experiments to support your analytical results. I suggest making some real examples (numerical examples) to show that your method really works on such NP-hard problems.  Otherwise, it is useless.

Just some minor comments:

1) In line 159,  the equation is wrong.

2) In line 165, a reference is missing.

3) In line 211," cannot" is better than "can not"

Regards.

Author Response

(The authors gave the same response as above.)

Round 2

Reviewer 1 Report

no more comments from my side. Please check the language of this manuscript carefully. There are still some minor errors.

Reviewer 2 Report

No problem.

Reviewer 3 Report

Dear Authors, 

I am still unsatisfied. I appreciated the efforts of making a real example with your algorithm.  However, your manuscript is now missing a  comparison with another algorithm.

Please compare it with the state-of-the-art algorithm in that example, and then I can be delighted with your revision. 

Regards.